# Genomic Analysis and Assessment of Melanin Synthesis in *Amorphotheca resinae* KUC3009

**DOI:** 10.3390/jof7040289

**Published:** 2021-04-12

**Authors:** Jeong-Joo Oh, Young Jun Kim, Jee Young Kim, Sun Lul Kwon, Changsu Lee, Myeong-Eun Lee, Jung Woo Kim, Gyu-Hyeok Kim

**Affiliations:** 1Division of Environmental Science & Ecological Engineering, College of Life Sciences & Biotechnology, Korea University, 145, Anam-ro, Seongbuk-gu, Seoul 02841, Korea; oheuy1027@korea.ac.kr (J.-J.O.); kjy4142@korea.ac.kr (J.Y.K.); sun-lul@korea.ac.kr (S.L.K.); 2Life Science and Biotechnology Department, Underwood Division, Underwood International College, Yonsei University, Seoul 03722, Korea; youngjn.kim@yonsei.ac.kr; 3Microbiology and Functionality Research Group, World Institute of Kimchi, Gwangju 61755, Korea; lckslick@gmail.com; 4Department of Biotechnology, College of Life Sciences & Biotechnology, Korea University, 145, Anam-ro, Seongbuk-gu, Seoul 02841, Korea; myeongeun88@gmail.com; 5Department of Biomedical Engineering, Sungkyunkwan University, Suwon, 2066 Seobu-ro, Jangan-gu, Suwon 16419, Korea; didch1789@gmail.com

**Keywords:** *Amorphotheca resinae*, fungal melanin, bioinformatics, melanin pigments

## Abstract

This study reports the draft genome of *Amorphotheca resinae* KUC30009, a fungal isolate with promising industrial-scale melanin production potential. The mechanisms for melanin or melanin-related pigment formation of this strain were examined through bioinformatic and biochemical strategies. The 30.11 Mb genome of *A. resinae* contains 9638 predicted genes. Genomic-based discovery analyses identified 14 biosynthetic gene clusters (BGCs) associated with secondary metabolite production. Moreover, genes encoding a specific type 1 polyketide synthase and 4-hydroxynaphthalene reductase were identified and predicted to produce intermediate metabolites of dihydroxy naphthalene (DHN)-melanin biosynthesis pathway, but not to DHN-melanin. These findings were further supported by the detection of increased flaviolin concentrations in mycelia and almost unchanged morphologies of the culture grown with tricyclazole. Apart from this, the formation of melanin in the culture filtrate appeared to depend on the laccase-like activity of multi-copper oxidases. Simultaneously, concentrations of nitrogen-containing sources decreased when the melanin formed in the media. Interestingly, melanin formation in the culture fluid was proportional to laccase-like activity. Based on these findings, we proposed novel strategies for the enhancement of melanin production in culture filtrates. Therefore, our study established a theoretical and methodological basis for synthesizing pigments from fungal isolates using genomic- and biochemical-based approaches.

## 1. Introduction

Contrary to their synthetic counterparts, the demand for natural pigments has been steadily increasing in recent years in response to global market shifts and consumer preferences [1,2]. Filamentous fungi have been gaining recognition as a potential microbial source of natural pigments; however, the industrial applications of these microorganisms are not as widespread as those involving algae or bacteria [3]. Fungi can potentially produce a wide range of pigments, such as carotenoids, *Monascus* pigments, and melanins [4,5,6,7,8], and can utilize a wide range of substrates, thus making the medium composition design and fermentation process more flexible [9].

However, fungal pigments are secondary metabolites, and most of their synthesis pathways and optimal production conditions remain largely unknown [10], which limits their optimization and widespread adoption in industrial applications. Fortunately, with the accumulation of high-throughput sequencing data, bioinformatics tools can be used to identify putative genes or gene clusters involved in metabolite production. Afterward, comprehensive prediction of the provisional biosynthetic pathways of pigments would be possible based on the identified putative genes or gene clusters [11]. Therefore, genomic studies could provide fundamental insights into the pathways associated with secondary metabolite synthesis in fungi, thus paving the way for their adoption in industrial-scale processes. We previously reported the promising capacity of the fungus *Amorphotheca resinae* KUC3009 to produce melanin at an industrial scale due to its antioxidant activity and high metal ion adsorption capability [8,12]. However, general information on the mechanisms of melanin or melanin-related pigment biosynthesis in *A*. *resinae* is not yet available. Therefore, understanding the mechanisms involved in melanin biosynthesis using bioinformatic tools would provide critical insights to optimize scalable pigment production using fungi.

Our study sequenced and assembled the whole genome of *A*. *resinae* KUC3009. After gene annotation, the putative genes and gene clusters involved in pigment formation were comprehensively investigated. Afterward, the involvement of putative genes or gene clusters was further corroborated with biochemical and molecular approaches. Based on these results, we proposed potential strategies for the optimization of scalable melanin production. Therefore, our study establishes a robust foundation for the production of secondary metabolites using *A*. *resinae*, as well as fundamental guidelines for future studies to improve upon.

## 2. Materials and Methods

### 2.1. Fungal Culture and DNA/RNA Extraction from Mycelia

*A. resinae* KUC3009 obtained from the Korea University Culture (KUC) collection was sub-cultured on sterilized cellophane membrane disks placed on potato dextrose agar for seven days at 25 °C. The mycelia on the cellophane membrane were then harvested, and their genomic DNA was extracted using a DNeasy plant mini kit (Qiagen, Valencia, CA, USA) according to the manufacturer’s instructions [13]. The quantity and quality of the extracted DNA were analyzed via the PicoGreen^®^ method and gel electrophoresis. Total RNA was also extracted from the same cultures using the RNeasy plant mini kit (Qiagen, CA, USA) according to the manufacturer’s instructions. The quantity and quality of the extracted RNA were assessed using an Agilent 2100 bioanalyzer (Agilent Technologies, Santa Clara, CA, USA) coupled with a DNA 1000 chip.

### 2.2. Library Construction and Whole-Genome Sequencing

Library construction and genome sequencing were carried out by Macrogen Co. Ltd. (Seoul, Korea). A DNA library with fragment sizes of approximately 20 kb was prepared with the SMRTbell template prep kit 1.0 (Pacific Biosciences, Menlo Park, CA, USA) and sequenced using the PacBio Sequel platform. Additionally, short reads were also constructed and sequenced with an Illumina HiSeq 4000 sequencer (San Diego, CA, USA). De novo assembly of PacBio sequence reads was performed using the hierarchical genome assembly process (HGAP) v4.0. After assembly, HiSeq reads were implemented to ensure a more accurate genome sequence using Pilon v1.21 [14]. An RNA-Seq library was also prepared and sequenced with an Illumina HiSeq 4000 system for high-quality genome annotation.

### 2.3. Genome Analysis, Annotation, and Phylogenetic Analyses

The completeness of the draft genome assembly was evaluated with BUSCO v3.0.2 using 1315 core genes of the Ascomycota dataset [15]. Genomic similarity comparisons between *A. resinae* and other strains were performed using orthoANI [16]. After the draft genome was analyzed, the gene locations were predicted using Maker (v2.31.8) [17]. tRNA and rRNA genes were predicted using tRNAscan (v1.4) and barrnap (v0.7, https://github.com/tseemann/barrnap, accessed on 1 November 2018), respectively [18]. The functions of the predicted genes were annotated using Protein BLAST+ (v2.6.0), after which the annotated genes were classified according to KOG analysis [19]. Gene clusters related to secondary metabolism were analyzed using antiSMASH Fungi v6.0 and secondary metabolite regions were identified using a “relaxed” strictness [20]. Lastly, Signal P5.0 was used to predict the presence of the signal peptide in translated products from the putative genes [21].

The internal transcribed spacer (ITS) region was selected for phylogenetic analysis of the selected fungus. The ITS region was amplified using the ITS1F (5′-CTT GGT CAT TTA GAG GAA GTA A-3′) and LR3 (5′-CCG TGT TTC AAG ACG GG-3′) primers. Polymerase chain reaction (PCR) was performed on a Bio-Rad MyCycler (Hercules, CA, USA) with the following protocol: initial denaturation at 95 °C for 5 min; 34 cycles at 95 °C (30 s), 55 °C (30 s), and 72 °C (30 s); final 5-min extension at 72 °C. DNA sequencing was carried out by Macrogen (Seoul, Korea) using the Sanger method with a 3730xl DNA analyzer (Life Technologies, Carlsbad, CA, USA). The ITS sequences were deposited in the GenBank database under accession numbers JN033458.2. The obtained ITS sequences were proofread and edited using reference sequences obtained from the GenBank database using MEGA v7.0, after which multiple alignments were conducted using MAFFT v7.130 [22,23]. The sequence alignments were manually modified when necessary. Additionally, a phylogenetic tree was constructed based on the ITS sequences of *A*. *resinae* and *Cladosporium*-like species using the “randomized accelerated maximum-likelihood” (RAxML) model coupled with the GTR+G evolution model and 1000 bootstrap replicates [24,25].

### 2.4. Tricyclazole Inhibition Assay and Measurement of Flaviolin in Mycelia

*A. resinae* was subcultured on potato dextrose agar for 10 days. Spores were then collected in 0.02% Tween-80, and their concentrations were adjusted to 1 × 10^6^ spores/mL. Afterward, cellophane membrane disks were inoculated with 10 µL of spore suspension and placed on potato dextrose agar (PDA) supplemented with different concentrations of tricyclazole ranging from 0 to 100 ug/mL. Colony sizes and colors were evaluated after 10 days via microscope imaging.

The secondary metabolites in the grown mycelia were extracted as described by Lisec et al. [26]. The mycelia on the cellophane membrane were harvested and flash-frozen with liquid nitrogen. The frozen mycelia were then ground, after which 100 mg of sample was transferred to 1.5 mL microcentrifuge tubes containing 1.0 mL of methanol to extract the metabolites. After adding 60 µL of a ribitol solution (0.2 mg/mL), the sample was incubated for 10 min at 70 °C with mild mixing. The incubated sample was centrifuged at 3500 rpm for 20 min, and the supernatant was transferred to a new tube. The supernatant was then mixed with 750 μL of chloroform and 1400 μL of distilled water, and the mixture was thoroughly vortexed. The mixed solution was centrifuged at 3500 rpm for 20 min, after which 150 μL of the supernatant was transferred to a new 1.5 mL tube. The extract was then fully dried and stored at −80 °C. Synthetic flaviolin was also prepared according to a previous study, and stock solutions were fully dried prior to analysis [27].

To derivatize both the synthetic and naturally occurring flaviolin in the mycelia, the samples were mixed with 40 μL methoxyamination reagent (20 mg/mL of methoxyamine hydrochloride dissolved in pyridine) and incubated for 2 h at 37 °C with constant mixing (150 rpm). After incubation, 70 μL of N-methyl-N-(trimethylsilyl)trifluoroacetamide (MSTFA) was added and incubated at 60 °C for 1 h. The concentrations of the resulting products were then determined via gas chromatography-mass spectrometry (GC–MS) (Agilent 6890 N GC equipped with a quadrupole Agilent 5973N MS spectrometer; Santa Clara, CA, USA). The samples were injected in splitless mode at 230 °C. The GC oven was held at an initial temperature of 80 °C for 2 min, which was increased to 325 °C at a 15 °C/min rate, then held at this temperature for 6 min. The separation process was carried out using a 25 m × 0.25 mm (inner diameter) × 0.25 μm DB–5MS UI capillary column with helium as the carrier gas at a 2 mL/min flow rate. Full scan acquisitions were performed over an m/z 50–800 range. Mass spectrometry was conducted at a 70 eV ionization energy, 230 °C source temperature, and 290 °C transfer line temperature. Automatic tuning of the instrument was conducted according to the manufacturer’s instructions.

### 2.5. Characterization of Melanin Production in the Culture Filtrate

The dry weights of the fungal biomass, melanin in the culture filtrate, and residual concentrations of glucose and total nitrogen (TN) were monitored throughout the cultivation process. Spore suspensions (10^6^ spores/mL) of the strain were prepared using the above-described methods. Afterward, 1 mL of the suspension was inoculated into individual flasks containing 100 mL of sterilized peptone yeast glucose (PYG; peptone: 10 g/L; yeast: 5 g/L; glucose: 20 g/L) media and PYG media supplemented with 1 mM CuSO_4_. The glucose concentration of the PYG media was adjusted from 5 g/L to 20 g/L. The inoculated culture media were maintained at a constant 150 rpm agitation on a rotary shaker at 27 °C. At each measurement point, the biomass and cell-free culture media were separated by centrifugation. The cells were weighed after filtering the sample through Whatman 1.2 μm glass fiber filters (Clifton, NJ, USA). The obtained supernatant was then acidified (pH 2) with 1 M HCl and incubated for 24 h at 21 °C to enable melanin formation. The newly formed melanin pellets were weighed after filtering the sample as described above. The filtered supernatant excluding the melanin pellets was used to measure the residual glucose and TN concentrations. The residual glucose concentrations were monitored using a high-performance liquid chromatography (HPLC) system (Shimadzu, Tokyo, Japan) equipped with a refractive index detector (RID-20A, Shimadzu, Tokyo, Japan) and an Aminex HPX-87H ion exchange column (300 mm × 7.8 mm) (Bio-Rad, Hercules, CA, USA) at a 0.5 mL/min flow rate. The mobile phase was a 5 mM H_2_SO_4_ solution prepared in deionized water. TN concentrations were determined using the HS-TN(CA)-L kit and an HS-1000PLUS water analyzer (Humas, Daejeon, Korea).

### 2.6. Laccase-Like Activity Assay

The laccase-like activity in the culture filtrate was determined as described by previous studies using 2,2′-azino-bis(3-ethylbenzthiazoline-6-sulfonate) (ABTS) as the substrate [28,29,30,31]. Briefly, 200 μL of culture filtrate was added to the enzyme assay solutions (100 mM acetate buffer, pH 5.0), resulting in a final concentration of 1 mM of ABTS. ABTS oxidation was monitored based on the increase in A_420_ (ε_420_ = 36,000 M^−1^ cm^−1^). One unit of enzyme activity was defined as the amount of enzyme required to oxidize 1 μmol of ABTS per minute at 25 °C.

### 2.7. Preparation of Synthetic Melanin and Its Characterization

The structural properties of the melanin synthesized using the culture filtrate supplemented with 1 mM CuSO_4_ and 10 mM L-3,4-dihydroxyphenylalanine (DOPA) solution were analyzed using Fourier-transform infrared spectroscopy (FT-IR). The synthesized melanin was precipitated by adjusting solution pH to 2.0 and then washed three times with deionized water. The melanin was air-dried and then mixed with KBr to prepare melanin-KBr pellets using a pellet press. The characteristic absorption spectrum of the pellet was collected using a Nicolet 6700 spectrometer (Thermo Scientific, Waltham, MA, USA). The measurement was carried out in a wavenumber range of 4000–650 cm^−1^ with a resolution of 4 cm^−1^. The number of scans per sample was 32. Automatic background and baseline corrections were applied to the obtained spectrum. The obtained spectrum was not further modified.

## 3. Results and Discussion

### 3.1. Taxonomy

A previous study reported that *A. resinae* strain KUC3009 cultures cause discoloration of chromated copper arsenate-treated wood [32]. This strain was formerly classified as *Cladosporium* sp.3, as sufficient molecular data were not available at the time. Therefore, our study reclassified this strain based on phylogenetic and morphological analyses. A BLAST search of the GenBank database using 526 bp ITS sequences of the selected strain indicated that the *Oidiodendron* and *Myxotrichum* genera were highly similar. Based on these similarities, we then sought to align 24 taxa, including *Byssoascus striatosporus,* as the outgroup taxon, after which a phylogenetic tree was constructed (Figure 1A). The isolate clustered closely with *A. resinae* isolates ATCC200942 and CBS406.68 with genetic similarities of 100% and 93%, respectively. Figure 1B,C shows light and electron microscopic images of the selected strains, respectively. Mycelia cultured on PDA were amorphous and showed strong pigmentation with a dark brown to black color, which was potentially due to the presence of a melanoid membrane. Furthermore, their subglobose or broadly ellipsoid to ovoid morphology with a 210 × 25 μm conidial size was consistent with previous reports [33]. Based on these observations, the isolate was identified as *A. resinae*.

### 3.2. Genome Sequencing

The genome of *A. resinae* KUC3009 was sequenced using a combination of PacBio and Illumina HiSeq reads with a 140x coverage. The genome assembly was approximately 30.11 Mb long and included 35 contigs with an average length of 183,235 bp (Table 1). The quality and completeness of the assembly were evaluated via BUSCO analysis. This analysis is routinely used to cross-analyze gene contents based on evolutionarily informed expectations of gene content [15]. Interestingly, 98.9% of the 1315 groups of genes required for the correct assembly of ascomycetes were present in *A. resinae* contigs, indicating that the *A. resinae* genome assembly was highly robust (BUSCO results are available in Appendix A). Only 0.4% and 0.7% of the gene groups were fragmented or missing, respectively.

The genomic features of *A. resinae* KUC3009 were compared with those of its sequenced relatives in the *Myxotrichaeae* family. The genome size of *A. resinae* KUC3009 (30.11 Mb) is similar to that of *A. resinae* ATCC 22711 (28.63 Mb) but smaller than that of *Oidiodendron maius* Zn (46.43 Mb) (Appendix A) [34]. For reference, the genome size of *A. resinae* was below the average genome size of common ascomycetes [35,36]. All species shared very similar genomic G+C content (i.e., approximately 47%). The average shared identity of three strains at the nucleic acid level was obtained with the OrthoANI calculator (Appendix A) [16]. The sequenced *A. resinae* strains were genetically closer to each other but were relatively distant from *O. maius*.

### 3.3. Genome Annotation

By utilizing several different gene predictors, we found that the *A. resinae* genome contained 9638 genes, 298 tRNAs, and 228 rRNAs (Table 1). The gene density was 3.20 genes per 10 kilobases (kb), and the predicted average protein size was 465 amino acids. The genes typically exhibited exons and introns with average lengths of 452 and 86 bp, respectively. Moreover, each gene contained an average of 3 exons. Additionally, the average length of the predicted proteins was 465 amino acids.

The predicted proteins from the genes were annotated and then functionally classified using eukaryotic orthologous group (KOG) analyses (Figure 2). The annotated proteins were classified into the following categories: “intracellular processes,” “metabolism,” “information storage and processing,” and “poorly characterized function” [37]. The results indicate that the proportion of the genes involved in the intracellular processes category was the highest, whereas the other categories were relatively insignificant. Among the metabolism category, the number of genes involved in lipid metabolism and the transport was slightly higher than that of the genes involved in carbohydrate metabolism and transport. *Cladosporium resinae*, the former name of *A. resinae*, is a common jet oil-deteriorating microorganism, thus highlighting its high capacity to degrade saturated hydrocarbons [38,39]. Consistent with previous studies, gene annotation identified homologs of alkane degradation enzymes, such as long-chain alcohol oxidase [40]. These results indicated that the genes and enzymes of *A. resinae* make this fungal species potentially well-suited for bioremediation.

### 3.4. Secondary Metabolite Biosynthesis Clusters

Fungi produce numerous secondary metabolites, which have a multitude of roles in cellular processes, such as transcription and development [41]. Many of these compounds have significant applications in the medical field (e.g., antibiotics and antitumor drugs), as well as in the agriculture sector (e.g., insecticides). Based on recent genome sequencing results, the ability of fungi to produce secondary metabolites has been largely underestimated, as the gene clusters associated with secondary metabolite biosynthesis are not expressed under laboratory growth conditions [10].

The biosynthesis gene clusters (BGCs) identified using antiSMASH v6.0 were classified according to their types, and BGCs assigned to the production of a certain product are described in Figure 3. Only 14 BGCs within five classifications were found in the genome of *A*. *resinae* KUC3009. The number of identified BGCs was significantly lower than that of other common ascomycetes. For reference, the number of predicted BGCs for *Penicillium* and *Aspergillus* ranges between 29 and 85 [36,42,43]. Here, only five clusters were associated with the production of certain metabolites: 1,3,6,8-tetrahydroxynaphthalene (1,3,6,8-THN), neurosporin A, brefeldin, phomopsins, and asperphenamate. One particular BGC in Type 3 polyketide synthase (T3-PKS) is predicted to produce phomopsins, and the gene exhibited a 100% similarity with that of *Phomopsis leptostromiformis*. Notably, phomopsins are a group of hexapeptide mycotoxins with potent antimitotic activity and, therefore, represent promising antitumor agents [44]. Beyond computational analysis, additional studies to prove the gene’s function are needed for using the strain as a source of antitumor production.

Our antiSMASH v6.0 analyses indicated that the genome of *A. resinae* retains T1-PKS BGC to produce 1,3,6,8-THN (Figure 3), which is the intermediate metabolite of the dihydroxy naphthalene (DHN)–melanin synthesis pathway (Figure 4A). This T1-PKS BGC of *A. resinae* exhibited a 100% similarity with that of *Bipolaris oryzae* and *Nodulisporium* sp. ATCC74245 (Figure 4B). These modules typically contain conserved domains that are comprised of acyl-carrier protein transacylase (SAT), β-ketoacyl synthase (KS), acyltransferase (AT), two acyl-carrier proteins (ACPs), and thioesterase (TE) (Figure 4C) [45,46].

The T1-PKS gene cluster associated with 1,3,6,8-THN synthesis is predicted to be involved in the scytalone synthesis but not with DHN-melanin synthesis (Figure 3). Through genome annotation analysis, we confirmed that complete putative tetra-hydroxynaphthalene reductase (4-HNR) coding genes were present in the genome, whereas scytalone dehydratase and tri-hydroxynaphthalene reductase (3-HNR) coding genes were not identified. For reference, 1,3,6,8-THN molecule goes through a series of catalytic reactions to form the final product (1,8-DHN) aided by 4-HNR, scytalone dehydratase, and 3-HNR, serially. Finally, the DHN molecules are polymerized to form DHN-melanin using oxidation enzymes, such as laccase or other phenol oxidases [47,48].

A series of biochemical evidence supports the existence of the T1-PKS gene associated with 1,3,6,8-THN and 4-HNR genes. Given that flaviolin is the auto-oxidative product of 1,3,6,8-THN, its detection in mycelia suggests that T1-PKS genes produce 1,3,6,8-THN [49]. Additionally, the amounts of flaviolin increased when the 4-HNR enzyme is inhibited [50]. We then sought to detect flaviolin and monitor its concentration after tricyclazole treatment to confirm whether our results were consistent with previous literature. GC–MS analyses confirmed the existence of flaviolin in mycelia cultured on PDA. Moreover, the flaviolin concentration in mycelia exhibited a 1.2-fold increase when cultured on PDA supplemented with 100 μg/mL of tricyclazole (Figure 4D).

The study with tricyclazole inhibition assays supported that 1,3,6,8-THN BGC from *A*. *resinae* is not involved in the DHN melanin formation. As tricyclazole inhibits DHN-melanin formation by repressing activity of 3-HNR as well as 4-HNR, fungal pigmentation with DHN-melanin was also inhibited when cultured with tricyclazole [50]. When cultured on tricyclazole-supplemented media, the effect of tricyclazole on pigment formation of *A*. *resinae* was poor, suggesting the pigmentation is not related to DHN-melanin synthesis. However, the gray-brown color of *A*. *resinae* cultures slightly changed to reddish-brown, which was likely due to an accumulation of shunt products (Figure 4E).

### 3.5. A. resinae KUC3009 Pigment Production in Culture Filtrate

We previously reported that *A*. *resinae* KUC3009 produced considerable amounts of melanin in the culture filtrate. The melanin synthesized by *A. resinae* KUC3009 in PYG media exhibited a high similarity with *Sepia* melanin in terms of their nitrogen content (approximately 7%) and indole-based chemical structures [8]. *Sepia* melanin is a typical type of eumelanin synthesized through the Raper–Mason pathway. L-tyrosine is a precursor, enzymatically oxidized by tyrosinase for melanin formation. Tyrosinase catalyzes the substrate using two consecutive activities of tyrosine hydroxylase and L-DOPA oxidase, resulting in the formation of L-dopaquinone [51,52,53]. Multi-copper oxidases, such as laccase, also can exhibit the activity of L-DOPA oxidase and be responsible for melanin formation [54].

The melanin derived from *A. resinae* slightly differed from *Sepia* melanin due to the following points; *Sepia* melanin results from the reaction of tyrosinase and L-tyrosine. However, L-tyrosine was not used in the preparation of the PYG media, and this strain cannot utilize the L-tyrosine as a substrate for melanin production (data not shown). Additionally, tyrosinase genes of *A*. *resinae* searched through gene annotation are predicted to be partial and intracellular (Appendix A), suggesting its activity in the culture filtrate would be insignificant in melanin formation. Instead, we focused on determining whether the pigments in the culture filtrate were formed through the laccase-like activity of multi-copper oxidase. Those enzymes are also known to possess a wide range of substrate specificity for melanin formation, and four putative complete genes predicted to have signal peptides were identified in the *A. resinae* genome (Appendix A).

*A. resinae* cultures were assessed based on the fungal biomass, melanin production, glucose concentrations, and laccase-like activity in culture filtrates (Figure 5). The fungal biomass increased prior to the stationary phase as the cells consumed the glucose in the culture medium. At a certain point of the stationary phase, both autolysis and melanin formation in the cultivation media occurred (Figure 5A–C). Interestingly, there was a surge in laccase-like activity prior to the melanin formation (Figure 5D–F). Interestingly, laccase-like activity was proportional to fungal biomass and melanin production in the cultivation media, and its surge exhibited a time lag as the initial glucose concentration in the media increased. The former was likely that enzyme production would be proportional to the fungal biomass, whereas the latter was possibly attributed to a glucose-suppressive effect, which has been reported in previous studies [55,56,57,58]. These results suggest a potential link between laccase-like activity in the culture filtrate and the melanin production of *A. resinae*.

The secreted enzymes can oxidize the nearby phenolic compounds to form melanins, whereas melanin can be formed via oxidation by enzymes released from the fungus during autolysis [48]. To unveil the exact mechanism of melanin formation in *A. resinae*, we induced melanin formation while keeping the cells alive by supplementing them with copper ions. As a result, both media’s fungal biomass increased with the incubation time, suggesting cells in both media were in the growth phase. However, the amount of melanin and laccase-like activity were significantly high only when supplemented with copper after 3 and 6 days of cultivation (Figure 6A,B). The results indicated that melanin formation in the culture filtrate could occur before the autolysis phase, suggesting secretion of the enzyme is a key determinant in melanin formation in media.

The substrate for melanin formation was not confirmed in this study. However, we discovered that the number of nitrogen sources in the metal-supplemented media was significantly decreased even though fungal biomass in copper-supplemented media was lower than that of normal PYG media (Figure 6A,B). Considering that the nitrogen content accounted for 7% of purified melanins derived from *A*. *resinae*, this suggests that *A*. *resinae* used nitrogen-containing substrates to synthesize melanin. Moreover, the purified melanin derived from *A. resinae* exhibited features of indolic moieties, which was likely due to the oxidation of indole-containing substrates derived from media components or metabolites derived from the strain. After removing mycelia, the culture filtrate supplemented with 1 mM CuSO_4_, exhibiting laccase-like activity, could synthesize the melanin using L-DOPA as a substrate. Figure 6C indicates the FT-IR spectrum of melanin synthesized using culture filtrate and L-DOPA, which exhibited a similar spectrum with that of L-DOPA melanin catalyzed by other laccases [59]. They commonly have the band at around 3400 cm^−1^ and peak at 1650 cm^−1^ due to the stretching vibrations of–OH and–NH_2_ groups and the vibrations of aromatic rings, respectively. These properties are commonly found within the melanin consisting of indole-based constituents [60].

Therefore, based on the above-described results, melanin production would likely be favored by enhanced laccase-like activity in the culture filtrate. Moreover, the selection of carbon sources and their concentrations should be prioritized to avoid glucose repression. Additionally, characterizing the effect of environmental factors on laccase-like activity is crucial.

## 4. Conclusions

Our study reported the draft genome of *A. resinae* and characterized its melanin synthesis mechanisms by combining bioinformatics and biochemical approaches. The genome size of *A. resinae* was relatively small than the average genome size of Ascomycetes. Moreover, only 14 BGCs were identified in the genome assembly, which contrasted with the higher BGC abundance of common ascomycetes. Genes encoding a specific T1-PKS and 4-HNR are predicted to produce intermediate metabolites of DHN-melanin biosynthesis but not proceed to DHN-melanin. These findings were further supported by the detection of increased flaviolin concentrations in mycelia and almost unchanged morphologies of the culture grown with tricyclazole. In the melanin formation in culture filtrates, it was observed that laccase-like activity and nitrogen sources are key determinants. Additionally, melanin formation in culture fluid was proportional to the laccase-like activity in the fluid. Melanin synthesis using L-DOPA and the culture filtrate exhibiting laccase-like activity were observed. Therefore, future studies should focus on enhancing laccase-like activity in culture filtrates to optimize industrial-scale melanin production using *A. resinae*.

## Figures and Tables

**Figure 1 jof-07-00289-f001:**
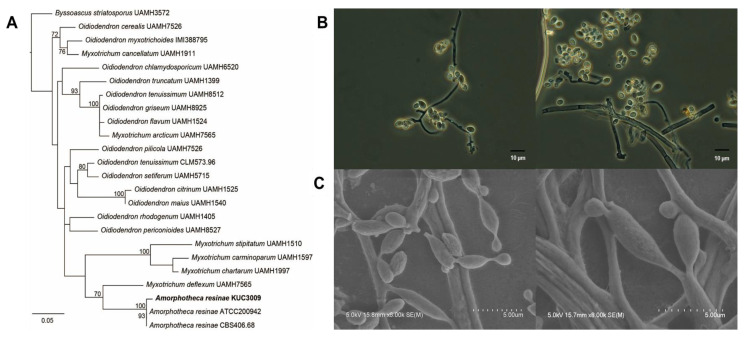
Molecular and morphological characterization of *A*. *resinae* KUC3009. (**A**) Phylogenetic tree based on internal transcribed spacer (ITS) sequence alignment generated by maximum-likelihood phylogenetic analysis. All bootstrap support values exceeded 70%. (**B**) Fungal culture imaging with light microscopy and (**C**) field emission-scanning electron microscopy.

**Figure 2 jof-07-00289-f002:**
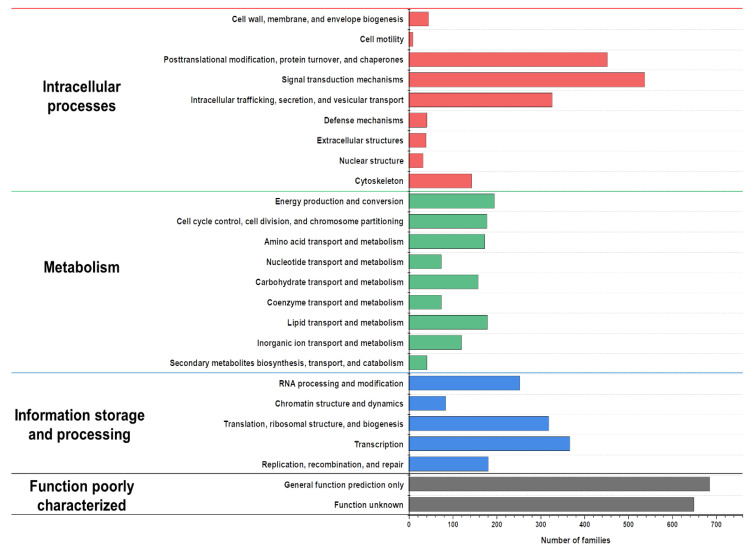
Eukaryotic orthologous group (KOG) distribution of predicted proteins from the *A*. *resinae* KUC3009 genome.

**Figure 3 jof-07-00289-f003:**
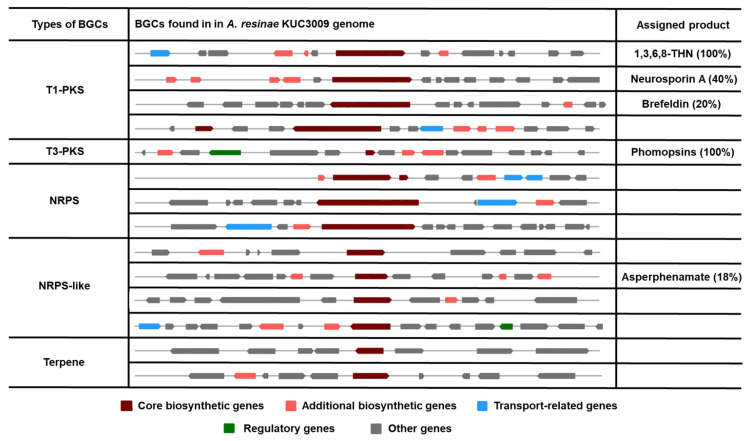
All identified biosynthetic gene clusters (BGCs) in the genome of *A. resinae* KUC3009 and their predicted assigned product. Values in parentheses indicate the similarity with a known cluster.

**Figure 4 jof-07-00289-f004:**
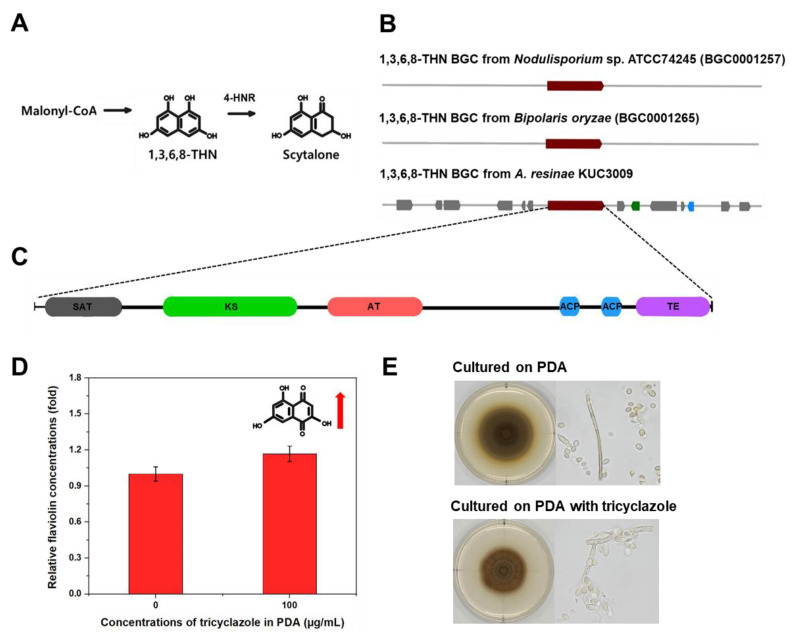
(**A**) Putative scytalone synthesis pathway of *A*. *resinae* KUC3009 (**B**) 1,3,6,8-THN BGC alignment between *A. resinae* and previously analyzed species. (**C**) Domain structure of the T1-PKS gene, consisting of an acyl-carrier protein transacylase (SAT) domain in gray, a β-ketoacyl synthase (KS) domain shown in green, an acyltransferase (AT) domain in pink, two acyl carrier protein (ACP) domains in blue, and a thioesterase (TE) domain in purple. (**D**) Comparison of flaviolin content in *A. resinae* mycelia cultured on PDA and tricyclazole-supplemented PDA (100 μg/mL). (**E**) Comparison of morphological properties of *A. resinae* cultured on PDA and tricyclazole-supplemented PDA (100 μg/mL).

**Figure 5 jof-07-00289-f005:**
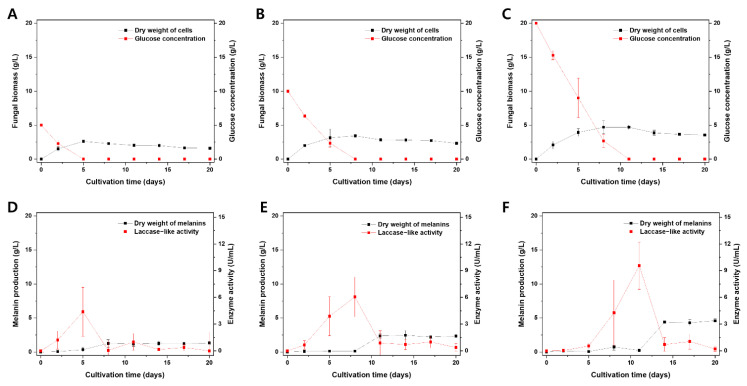
Effect of initial glucose concentration on the *A*. *resinae* cultivation profile. Fungal biomass and glucose concentration of media when cultivated at an initial glucose concentration of (**A**) 5 g/L, (**B**) 10 g/L, and (**C**) 20 g/L. Melanin production and laccase-like activity in culture filtrate when cultivated at an initial glucose concentration of (**D**) 5 g/L, (**E**) 10 g/L, and (**F**) 20 g/L.

**Figure 6 jof-07-00289-f006:**
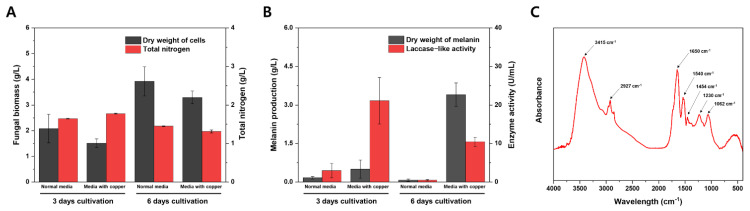
Effect of copper addition on *A*. *resinae* cultivation profile. (**A**) Fungal biomass and total nitrogen concentration of media. (**B**) Melanin production and laccase-like activity in the culture filtrate. (**C**) FT-IR spectrum of melanin synthesized using the culture filtrate and L-DOPA.

**Table 1 jof-07-00289-t001:** Genome assembly statistics for *A. resinae* KUC3009.

Assembly Statistics	Value	Gene Statistics	Value
Number of contigs	35	Number of genes	9638
Length of the largest contig	3,753,173	Number of tRNAs	298
Average contig length	860,288	Number of rRNAs	228
Total contig length	30,110,100	Protein length (amino acids, median)	465
N50	2,338,627	Exon length (bp, median)	452
Genome coverage	140×	Intron length (bp, median)	86
G+C content (%)	47.5	Average exon number per gene	3.0

## Data Availability

Not applicable.

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
