# Peer review of "Genomic Analysis and Assessment of Melanin Synthesis in Amorphotheca resinae KUC3009"

_jof, 2021, doi:10.3390/jof7040289_

Round 1

Reviewer 1 Report

This manuscript reports the whole sequencing and assembling of the A. resinae KUC3009 genome. The putative genes and gene clusters involved in melanin formation are investigated. Afterward, the involvement of putative genes or gene clusters is corroborated with molecular approaches. Based on results, the article proposed some strategies for the optimization of melanin production. The study also establishes a basis for the production of other interesting secondary metabolites aside melanin using A. resinae,and some relevant guidelines for future studies.

This is correct and inside the scope of the Journal. I have no criticisms on the part of the article concerning the draft genome of Amorphotheca resinae KUC30009.

However, the study is presented by a group with previous contributions for using this fungus Amorphotheca resinae KUC3009 to produce melanin at an industrial scale. However, general information on the mechanisms of melanin was only partially known, and the group is treating to contribute to the understanding of the mechanisms involved in melanin biosynthesis using bioinformatic tools. This is plausible, but there are important points to address. 

The main concern of this study is related to the melanin type formed, or in other words to the melanin structure. Briefly, in fungi, two types of melanin are possible: Dopa-melanin and DHN-melanin. In this paper, according to the genome studies, the nature of the melanin is undefined, and so, the involvement of a laccase or a tyrosinase is unknown. The abstract suggests that authors are thinking about DHN-melanin. They describe genes encoding a specific type 1 polyketide synthase and 4-hydroxynaphthalene reductase to predict production of intermediates for the DHN-melanin biosynthesis pathway. However, also in the abstract, it is stated that melanin formation was largely dependent on nitrogen-containing sources in the media. DHN-melanin does not incorporate nitrogen. Dopa melanin does. Figure S1 suggests that dopa-melanin is not formed, as PTCA is not detected in the fungal pigment, but it is in the cuttlefish melanin (proven DOPA-melanin). Experiments with flaviolin are also congruent with this idea, as well as the scheme showed at Figure 4.

However, at Fig. 4E, comparison of morphological properties of A. resinae cultured on PDA and tricyclazole-supplemented PDA (100 μg/mL), inhibition is not visible suggesting that melanin is not formed by the scytalone pathway. Then, section 3.5.2 is surprising. Discussion about extracellular melanin production focuses on the Raper-Mason pathway of eumelanin formation through formation of dihydroxyindoles using L-tyrosine as precursor. But Tyrosinase is just mentioned at lines 352 and 442. The last one is referred to Table S4, but this Table does not show any tyrosinase, just four putative laccases. This is unacceptable.

Lines 399-406: “The substrate for melanin formation was not confirmed in this study. However, we discovered that the amount of nitrogen sources in the metal-supplemented media was significantly decreased even though fungal biomass in copper-supplemented media was lower than that of normal PYG media. Considering that the nitrogen content accounted for 7% of purified melanins derived from A. resinae, this suggests that A. resinae used nitrogen-containing substrates to synthesize melanin. Moreover, the purified melanin de-404 rived from A. resinae exhibited features of indolic moieties, which was likely due to the  oxidation of indole-containing substrates derived from peptone or its hydrolysates.

The dependence of pigment formation by nitrogen sources, the poor effect of tryclazole on pigment formation and the nitrogen content in the melanin pigment should be discussed. Some of these data do not reconcile with the DHN-melanin nature of the pigment. Anyway, section 3.5.2 and lines 399-406 should be clearly re-written, as eumelanin coming from L-tyrosine does not seem to be the pigment formed by this fungus. Otherwise, this should be proven.

Corcerning these points, I have the following comments

  • The nature of the enzyme involved in melanin formation should be determined. Note that some multipotent copper-oxidases with laccase and tyrosinase activities have been described (Sanchez-Amat, et al., 2001). Molecular cloning and functional characterization of a unique multipotent polyphenol oxidase from Marinomonas mediterranea. BBA, 1547(1), 104-116
  • Some fungi, such as Cryptococcus neoformans, is able to form both, DHN-melanin and Dopa-melanin. This strain contains tyrosinase (polyphenol oxidase) and laccase activity. Could this fungus also contain both types of melanins to account for all the conflicting data presented.

Other minor points

Line 379-381: There are two mechanisms by which melanin formation may occur in A. resinae culture filtrates: (1) external secretion of phenol oxidases such as laccases and (2) secretion of phenolic compounds into the external environment. This is wrong, just delete.

Laccase activity. The laccase activity in the culture filtrate was determined as described by previous 190 studies using 2,2′-azino-bis(3-ethylbenzthiazoline-6-sulfonate) (ABTS) as the substrate  This substrate shows a low specificity and it is oxidized by a series of oxidants an enzymes, so that the laccase activity determined is not for sure. Alternative substrates/assays for laccase should be tried.

Lines 395-on: "Copper addition upregulated the expression of putative laccase gene 2, whereas that of putative laccase gene 1 remained generally low or undetected". How the expression of both laccase genes are discriminated? Laccase assay is not useful for that. Copper is essential for laccases and tyrosinases

Author Response

We appreciate your comments. The comments were really helpful for improvement of the manuscript. Revisions were marked in red in the revised manuscript, and comments were replied by each point.

Reviewer 2 Report

This manuscript presents new insights in the biosynthesis of melanin pigment pf Amorphotheca resinae KUC3009 based on genomic analysis. Introduction is clear and concise, with the aim of the study clearly stated. Genome and phylogeny analysis are well described, and solid literature data was provided for sustaining these methods. The whole study is well structured. The materials and methods section are presented in detail; therefore, the protocols could be followed by other readers. Melanin analysis methods are adequate.

The discussion is clear and conclusions are based on solid results.

Some minor issues are:

Line 185: please add the chromatographic column size

Line 230, Figure 1B: please add scale bar for this figure

Author Response

(The authors gave the same response as above.)

Round 2

Reviewer 1 Report

Dear Sirs,

I have examined the reply letter and the amended version of the manuscript. I have to say that  manuscript has been modified in the correct way, and the initial confusion about the nature of the melanin has been reduced. I feel that authors decided that the melanin they are dealing with are Dopa-melanin rather than DHN-melanin. It is clear that they discard the option that Amorphotheca resinae KUC3009 would be able to form both types of melanins. I suggested that possibility  in my previous review(see for instance literature about C. neoformans), but they ignored that possibility.

In that way, they re-wrote some parts of the manuscript for adapting data for compatibility to dopa-melanin. For instance, the chromatogram showing the absence of PTCA as a marker of dopa-melanin disappeared, as PTCA should be present for proving dopa-melanin. An FT-IR spectrum is now added, but this spectrum is not totally convincing. In turn, the first version of the manuscript described the detection of four loci for possible laccase enzymes, and in the new version, these four loci are now putative tyrosinases, again looking for consistency with dopa-melanin. Ok, I could assume those adaptations, but overall manuscript should be consistent with the nature of the melanin formed.

This is not the case. Therefore, I suggest some new modifications before the manuscript would be accepted.

Line 31: Abstract “specific substrates for melanin formation have not been identified yet”. This statement is not needed at the abstract. In the same way, the sentence at lines 336-338 should be deleted. These statements are  not consistent with the fact that  L-dopa has been identified as precursor of melanin (this point is stated three times at the manuscript, see lines 186, 386 and 418, conclusion). It is not possible formation of L-dopa and the subsequent dopa-melanin without the initial hydroxylation of L-tyrosine to L-dopa by a tyrosinase or at least a phenol oxidase/catechol oxidase.

According to that, it is odd and inconsistent the maintenance of a part of Figure 4 at thew amended manuscript. That figure shows the pathway for DHN-melanin, but DHN-melanin is not formed. Authors propose that the fungus synthesizes Dopa-melanin, bot DHN-melanin, but they include the pathway leading to DHN-melanin and they do not include the Raper-Mason pathway leading to dopa-melanin.  I recommend adding some general review about melanins to references (i.e. http://dx.doi.org/10.1155/2014/498276, see figure 5).

Line 392: In agreement with that, the indolic structures are formed from L-dopa through formation of dopachrome and dihydroxyindoles, but tryptophan is not necessary. The discussion at this line is unnecessary and misleading. The Raper-Mason pathway is never mentioned in the manuscript, and it should be to understand some of the proposed data. I think the authors are familiar with bioinformatics and fungal genetics, but they are not familiar with melanogenesis.

I think that authors have tried to grow the fungus at PYG media using L-tyrosine, but they failed. This does not mean that L-tyrosine is not the precursor of melanin . I think that the transformation of L-tyrosine to L-dopa is difficult and tricky (I cannot explain this point just now, it would be long to account for this point, but PYG could not be the best media for that type of experiment). In the same way, the manuscript would need to include assay for tyrosinase, or at least for dopa oxidase in order to detect possible tyrosinase activity. Anyway, the paper contains a long and valuable part on fungal genetics, and I would recommend the definitive acceptance of the paper without including these new data.
